


# A New Vision of the Adriatic Dense Water Future under Extreme Warming

Cléa Denamiel[1,2], Iva Tojčić[1,3], and Petra Pranić[4]

[1]Ruđer Bošković Institute, Division for Marine and Environmental Research, Bijenička cesta 54, 10000 Zagreb, Croatia

[2]Institute for Adriatic Crops and Karst Reclamation, Put Duilova 11, 21000 Split, Croatia

[3]Faculty of Science and Mathematics of Split, Ruđera Boškovića 33, 21000 Split, Croatia

[4] Institute of Oceanography and Fisheries, Šetalište I. Meštrovića 63, 21000 Split, Croatia

*Correspondence to*: Cléa Denamiel (cdenami@irb.hr)

**Abstract.** We use the Adriatic Sea and Coast (AdriSC) kilometer-scale atmosphere-ocean model to assess the impact of a far-future extreme warming scenario on the formation, spreading, and accumulation of both the North Adriatic dense Water (NAddW) over the entire basin, including the Jabuka Pit accumulation site, and the Adriatic Deep-Water (AdDW) over the Southern Adriatic Pit (SAP). Our key findings differ from previous studies that used coarser Mediterranean climate models and did not update the thresholds for dense and deep- water definitions to account for the far-future background density
changes caused by warmer sea surface temperatures. We show that surface buoyancy losses at NAddW generation sites, driven by evaporation, are expected to increase by 15% under extreme warming, despite a 25% reduction in the intensity and spatial extent of Bora winds. As a result, future NAddW formation will remain similar to present conditions. However, the volume of dense water in the Jabuka Pit will decrease due to the increased far-future stratification. Additionally, dense water transport between the Jabuka Pit and the deepest part of the SAP will stop, as future NAddW will be lighter than the AdDW.
Regarding Ionian-Adriatic exchanges, extreme warming will not affect the impact of the Bimodal Oscillation System on the Adriatic salinity variability, but future AdDW dynamics will be determined by density changes in the northern Ionian Sea. Our findings highlight the complexity of climate change impacts on Adriatic atmosphere-ocean processes and the importance of high-resolution models for more accurate far-future projections in the Adriatic Sea.

## 1 Introduction

Dense waters, generated by extreme air-sea buoyancy losses, play a crucial role in the health and functioning of the oceans worldwide. These waters drive local and basin-wide thermohaline circulation (Broecker, 1991; Rahmstorf, 2002), ventilate deep ocean layers to support marine life, and facilitate the global carbon cycle (Emerson et al., 2004; Gruber, 2011). They transport essential nutrients (e.g., nitrogen, phosphorus, and iron) from the surface to deeper ocean layers, supporting primary production and marine ecosystems, and promoting the growth of phytoplankton and other organisms at the base of
the marine food web (Martin and Fitzwater, 1988; Boyd et al., 2007). Additionally, they drive vertical mixing and upwelling, which enhances biological productivity and biodiversity in surface waters (Vélez-Belchi et al., 2018; Doney et al., 2012). These processes influence regional and global climate patterns by transporting heat, moisture, and carbon dioxide across



ocean basins (Rahmstorf et al., 2015; IPCC, 2019). However, with ongoing and future global warming, increased ocean stratification will inhibit the transport of heat, oxygen, and carbon dioxide from the surface to deeper layers, intensifying

ocean acidification and impacting the marine food chain (Li et al., 2020).

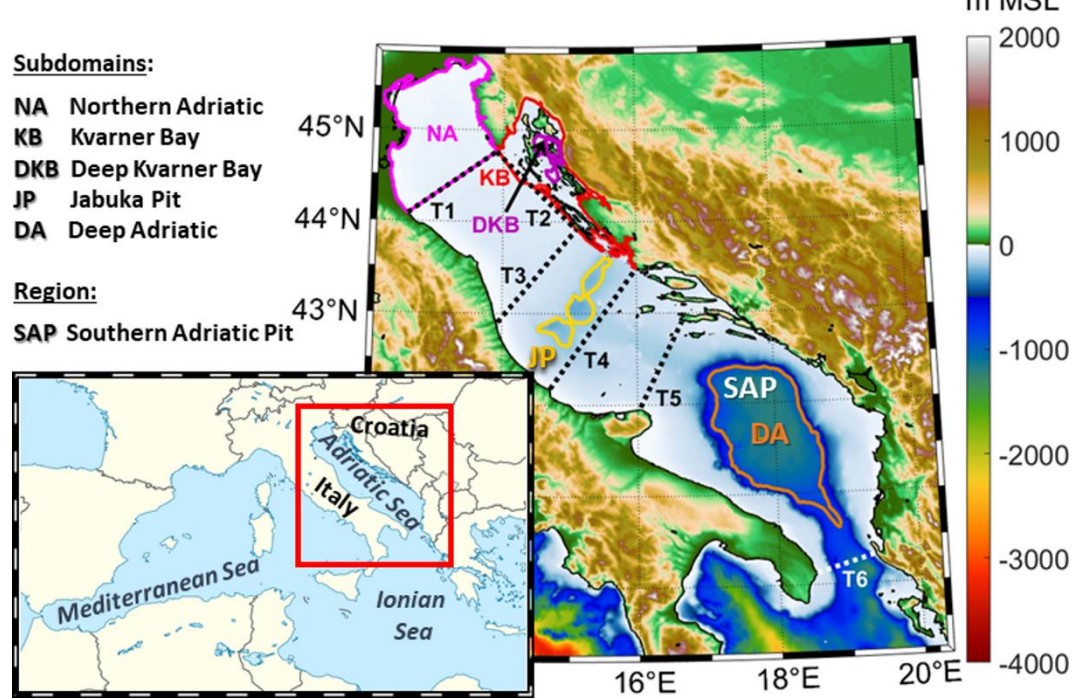

**Figure 1. Topo-bathymetry of the AdriSC climate model with the locations of the 5 subdomains (colored polygons) and 6 transects (dotted black and white lines) used in the study.**

In the Mediterranean Sea, the densest waters are formed in the northern Adriatic Sea (Fig. 1) during extreme winter

windstorms known as Bora events, which produce hurricane-strength gusts up to 50 m/s (Belušić and Klaić, 2004) and lead to significant sea surface cooling (e.g., Ličer et al., 2016). The dynamical properties of the Northern Adriatic Dense Water (NAddW; Zore-Armanda, 1963) in the present climate have been extensively studied over the last 60 years, as summarized by Vilibić et al. (2023). NAddW formation occurs within the shallow Northern Adriatic shelf (Fig. 1, NA subdomain) and the Kvarner Bay (Fig. 1, KB subdomain) during strong surface heat and freshwater losses between December and March.

The deepest part of the Kvarner Bay (Fig. 1, DKB subdomain) also acts as a dense water collector. NAddW then spreads southward along the western Adriatic coast and is partially collected within the Jabuka Pit (Fig. 1, JP subdomain) and the Southern Adriatic Pit (SAP; Fig. 1). In the SAP, the Adriatic Deep Water (AdDW) is generated through open ocean convection, strongly preconditioned by the presence of a permanent cyclonic gyre. Finally, the remaining NAddW and AdDW exit the Adriatic basin through the Strait of Otranto towards the northern Ionian Sea.

NAddW and AdDW are the main sources of Eastern Mediterranean Deep Water (EMDW; Pollack, 1951; Malanotte-Rizzoli et al., 1997), playing a significant role in sustaining the Mediterranean overturning circulation (Li and Tanhua, 2020) and



shaping the biogeochemical processes and ecosystem dynamics of the Eastern Mediterranean Sea (Herut et al., 2016; Thingstad et al., 2005; Rahav and Herut, 2016). However, the impact of climate change on their dynamical properties has not been thoroughly assessed. Parras-Berrocal et al. (2023) recently studied the impact of climate change on dense water

formation in the Eastern Mediterranean, but their results were averaged over the entire Adriatic Sea. The Regional Climate Model (RCM) they used also has coarse resolutions – 25 km in the atmosphere and about 15 km in the ocean – insufficient to represent the known NAddW dynamics accurately. Indeed, Denamiel et al. (2021a) and Pranić et al. (2023) have shown that only nonhydrostatic kilometer-scale atmospheric models and ocean models with at least 1 km resolution can properly reproduce the dense water dynamics within the Adriatic basin.

The kilometer-scale Adriatic Sea and Coast (AdriSC) climate model (Denamiel et al., 2019) is thus used in this study. The abilities of the AdriSC model to simulate both extreme Bora events in the atmosphere and dense water dynamics within the Adriatic basin have been assessed in the present climate, with many studies demonstrating the added value of such a kilometer-scale atmosphere-ocean climate approach (Denamiel et al., 2020a, b, 2021a, b, 2022; Pranić et al., 2021, 2023, 2024; Tojčić et al., 2023, 2024). Consequently, the present study focuses on understanding and analyzing in detail the far-

future impacts of an extreme warming scenario on the atmosphere-ocean processes driving the NAddW and AdDW dynamics. The article is structured as follows. The AdriSC model and the methods used for the analyses are described in Section 2, while the impacts of climate change on the Bora events, Adriatic dense water dynamics, and Ionian-Adriatic exchanges are assessed and discussed in Section 3. Finally, conclusions about the main findings of the study are presented in Section 4.

## 75   2 Model and Methods

### 2.1 Adriatic Sea and Coast (AdriSC) Model

#### 2.1.1 AdriSC Model Setup

The kilometer-scale Adriatic Sea and Coast (AdriSC) climate model (Denamiel et al., 2019) has been developed to represent the atmospheric and oceanic circulation over the Adriatic basin with greater accuracy than the available Mediterranean

Regional Climate Models (RCMs). It is based on the Coupled Ocean–Atmosphere–Wave–Sediment Transport (COAWST) modeling system (Warner et al., 2010), which dynamically couples the Weather Research and Forecasting (WRF; Skamarock et al., 2005) atmospheric model and the Regional Ocean Modeling System (ROMS; Shchepetkin and McWilliams, 2009). As illustrated in Fig. 2 (top panel), two nested grids of 15-km and 3-km resolution are used in the WRF model, and two nested grids of 3-km and 1-km resolution are used in ROMS. Vertically, terrain-following coordinates are

used with 58 levels refined in the surface layer for the atmosphere (Laprise, 1992), and 35 levels refined near both the sea surface and bottom floor for the ocean (Shchepetkin and McWilliams, 2009).



The AdriSC modeling suite is installed and fully tested on the European Centre for Medium-Range Weather Forecasts (ECMWF) high-performance computing facilities. More details on the AdriSC setup can be found in Denamiel et al. (2019, 2021b) and Pranić et al. (2021).

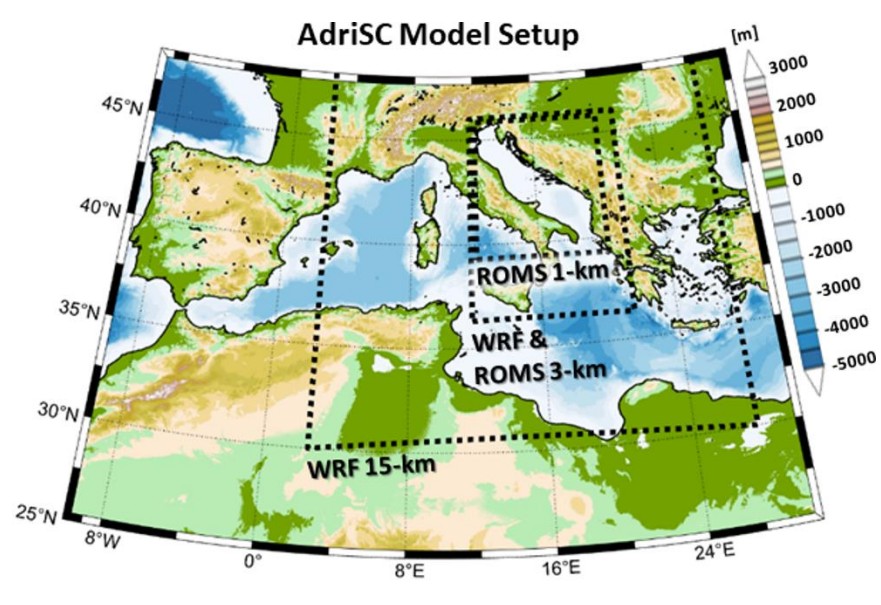

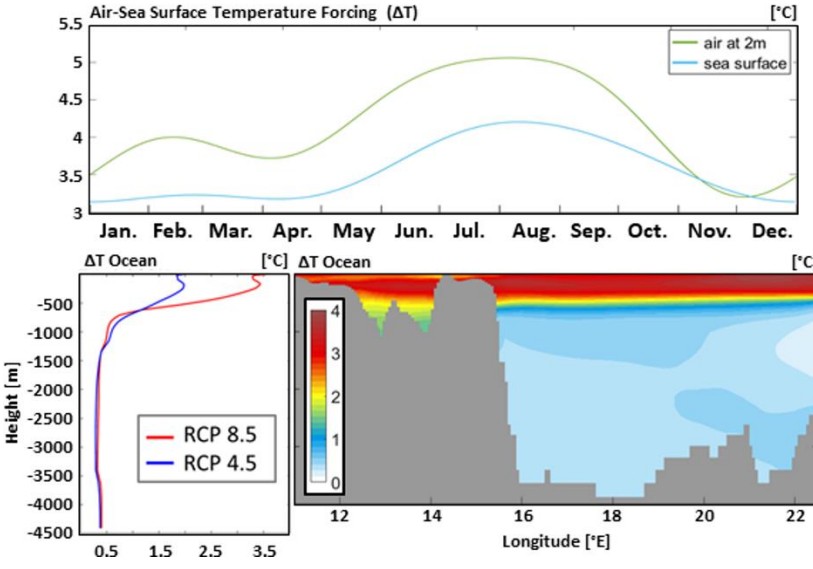

**Figure 2. Spatial coverage and horizontal resolution of the different grids used in the AdriSC climate model (top panel) and Pseudo Global Warming temperature forcing imposed in the AdriSC extreme warming simulation (bottom panels).**






### 2.1.2 Pseudo Global Warming (PGW) Approach

In this study, the impact of climate change is assessed with two 31-year-long AdriSC climate simulations: a historical run for
the 1987-2017 period and a far-future extreme warming run (2070-2100 period) based on the Representative Concentration
Pathway (RCP) 8.5 (hereafter RCP 8.5 simulation). For the historical run, the initial and boundary conditions are provided to
the WRF 15-km model by the 6-hourly ERA-Interim reanalysis fields at 0.75° resolution (Dee et al., 2011) and to the ROMS
3-km model by the Mediterranean Forecasting System (MFS) MEDSEA reanalysis at 1/16° resolution (Simoncelli et al.,
2016, 2019). The AdriSC historical climate run has already been successfully evaluated (Denamiel et al., 2021b; Pranić et
al., 2021) and proven to reproduce the known Adriatic multi-decadal dense water dynamics (Pranić et al., 2024).

For the RCP 8.5 run, the Pseudo Global Warming (PGW) methodology is used to address both the relative slowness of the
AdriSC model (i.e., a month of results produced per day) and the low temporal and spatial resolutions (i.e., few vertical
levels for daily or monthly results) of the RCMs available to force the AdriSC WRF 15-km and ROMS 3-km models. The
principle of the PGW simulations (Schär et al., 1996; Denamiel et al., 2020a) is to impose an additional climatological
change (e.g., temperature change $\Delta T$ ) to the reanalysis used to force the historical run. Here, the results of the LMDZ4-
NEMOMED8 RCM (Hourdin et al., 2006; Beuvier et al., 2010) are used to produce the PGW forcing – see Denamiel et al.
(2020a) for a detailed description. As illustrated in Fig. 2 (bottom panels), the PGW temperature forcing imposed in the
AdriSC RCP 8.5 simulation is about 1 °C warmer for the air than the sea at the surface. It is also below 0.5 °C in the ocean
for all depths below 1000 m but can reach up to 3.5 °C between the surface and 200 m depth for the RCP 8.5 scenario. These
strong vertical gradients of temperature imposed on the ocean reanalysis are thus expected to impact the density of the
Adriatic Sea, which will be far lower in the shallow areas of the basin (e.g., NA and KB subdomains) than in its deepest part
(e.g., DA subdomain) in the RCP 8.5 simulation.

### 2.2. Methods

### 2.2.1 Bora Events

To understand the impact of climate change on the air-sea interactions driving the NAddW formation, the atmospheric
results – derived from the AdriSC WRF 3-km daily fields – are only examined over the northern Adriatic Sea (for latitudes
above 43 °N) during extreme Bora events defined for wind speeds at 10 m greater than 13 m/s (i.e., gale force winds).
Firstly, the validity of this simple criterion is demonstrated by analyzing the median monthly wind speeds at 10 m (≥ 13 m/s)
during the 31 years of the historical simulation and comparing the obtained results with the known Bora jet dynamics.
Secondly, the impact of climate change on the selected Bora winds is assessed with spatial plots of the climate adjustments
(in percent) defined, during the 31 years of the simulations, as the difference between RCP 8.5 and historical median
monthly wind speeds divided by the historical median monthly wind speeds. Then, monthly climatologies are presented as
time series of the median, 25[th] and 75[th] percentiles of the historical and RCP 8.5 results for 8 different variables: horizontal
wind transport at 10 m, accumulated surface buoyancy loss, total, sensible and latent heat fluxes, air minus sea saturation



specific humidity (SAT), relative humidity at 2 m and fresh water fluxes (see Appendix A for the mathematical definition of the variables). Finally, the results are summarized with a box plot presenting, for the 8 variables, the climate adjustments (in percent) defined as the difference between the RCP 8.5 and historical monthly results divided by the historical monthly results during the December to March (DJFM) period when the NAddW is known to be formed.

### 2.2.2 Dense Water Dynamics

In this study, all ocean variables are derived from the daily AdriSC ROMS 1-km fields and the potential density anomalies (PDAs, $\sigma$), as well as the thermal expansion and haline contraction coefficients, are calculated with the equation of state introduced by McDougall, Wright, Jackett, and Feistel (MWJF; Levitus et al., 1994a, 1994b; Dukowicz, 2000). Under the present climate, the NAddW is characterized by densities $\sigma \geq 29.2$ kg/m³ (Mantziafou and Lascaratos, 2008). However, the NAddW is formed in the shallowest part of the Adriatic Sea where a strong change in background density is imposed by the

PGW forcing (Fig. 2, bottom panels). Therefore, this threshold cannot be used to analyze the RCP 8.5 simulation. The NAddW is known to exit the Adriatic basin along the shallow western shelf of the Strait of Otranto (Fig. 1, transect T6). In Figure 3, the historical and RCP 8.5 PDAs are presented as spatial plots of median (over the 31 years of the simulations) along the T6 transect and probability density functions – calculated with a kernel-smoothing method (Bowman and Azzalini, 1997) and evaluated for 100 equally spaced points – derived along the Strait of Otranto, at the bottom of the western shelf

(Fig. 3, top panel, black box). This analysis reveals that a density of $\sigma = 29.2$ kg/m³ is obtained for the 97th percentile of the historical PDAs, which corresponds to $\sigma = 28.4$ kg/m³ in the RCP 8.5 simulation (Fig. 3) and defines the criterion used to identify the far-future NAddW. The SAP which is the deepest area of the Adriatic Sea is also excluded from the following analyses.

The impact of climate change is first assessed for the Dense Water Height (DWH; see Appendix A for the mathematical
definition) as spatial plots of the median (over the 31 years of the simulations) of both the historical monthly maximums and the climate adjustments (in percent) defined as the difference between RCP 8.5 and historical monthly maximums divided by the historical monthly maximums. Then, daily climatologies are presented as the median, 25th and 75th percentiles of the historical and RCP 8.5 results for 3 different variables: Dense Water Volume (DWV) and Stratification Index (SI) over 4 different subdomains (NA, KB, DKB, and JP; Fig. 1) identical to those used in Pranić et al. (2024) as well as outward (i.e.,
exiting the Adriatic basin) NAddW mass transport along 5 different transects (T1 to T5; Fig. 1) defined along the known dense water pathways (see Appendix A for the mathematical definition of the 3 variables). Finally, the results are both summarized with box plots of the climate adjustments (in percent) defined as the difference between the RCP 8.5 and historical daily results divided by the historical daily results during DJFM, and further analyzed with historical and RCP 8.5 PDA pycnoclines along the T3 to T5 transects and within the JP subdomain. An animation of DWH over the Adriatic Sea is
also provided for the RCP 8.5 simulation (Movie S1).



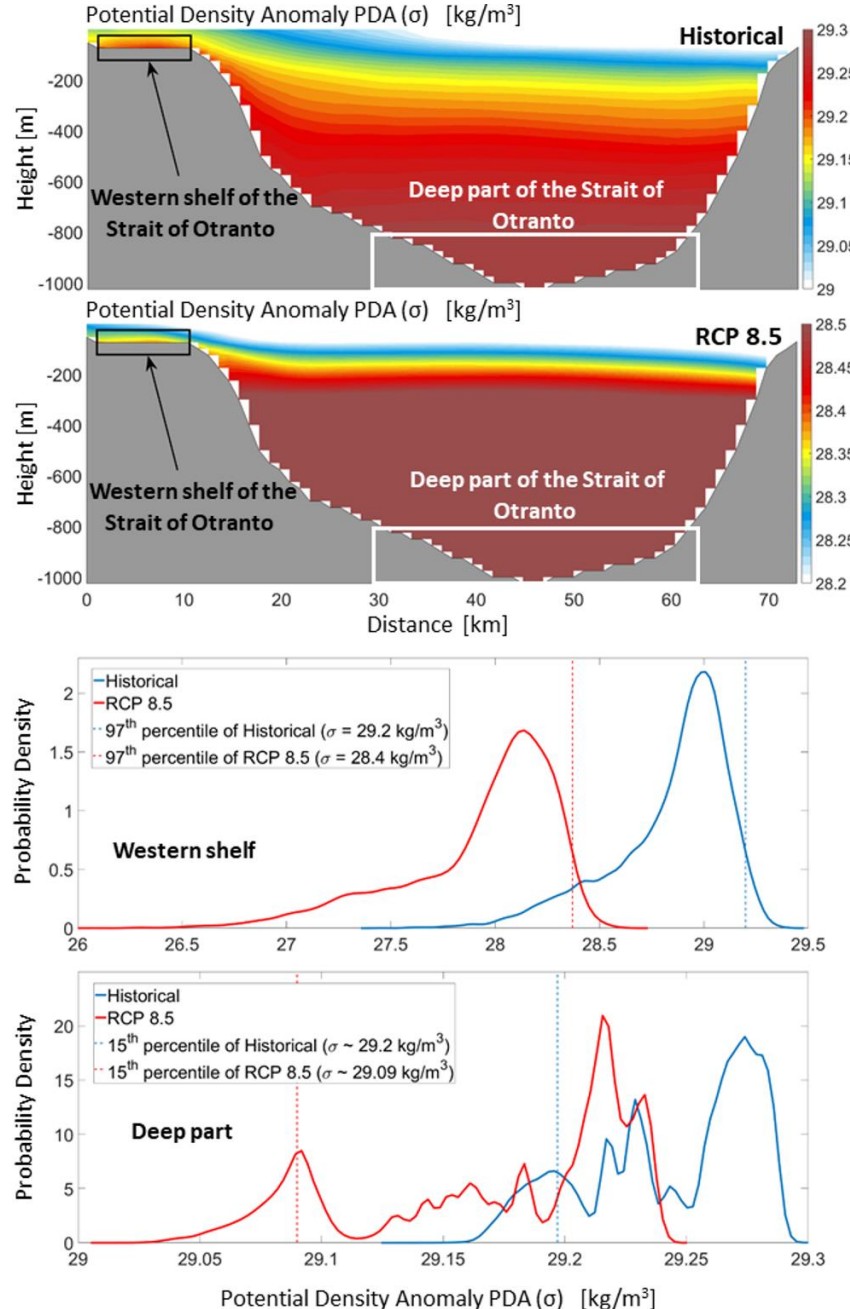

**Figure 3. Spatial plots of the vertical transect along the Strait of Otranto for the median of the PDAs over the 31-year historical and RCP 8.5 simulations (top panels) and historical and RCP 8.5 PDA probability density functions at the bottom of the western shelf (black box) and for depths greater than 800 m (white box) along the Strait of Otranto.**



### 2.2.3 Ionian-Adriatic Exchanges

Due to the strong density gradients between the shallow and deep areas of the Adriatic Sea under the RCP 8.5 scenario, most of the AdDW exchanges within the SAP are expected to occur with the northern Ionian Sea, a deep basin with depths greater than 3500 m. An analysis of the PDAs for depths below 800 m along the Strait of Otranto (Fig. 3, top panel, white box)

reveals that the present climate deep-water density threshold $\sigma = 29.2$ kg/m³ (Gačić et al., 2001) is obtained for the 15th percentile in the historical simulation, which corresponds to $\sigma = 29.09$ kg/m³ in the RCP 8.5 simulation (Fig. 3). This defines the criterion used to identify the far-future AdDW.

Empirical Orthogonal Functions (EOFs) are used to compare, in space and time, the most important variability patterns in the Adriatic and northern Ionian seas for both historical and RCP 8.5 simulations. Denamiel et al. (2022) demonstrated that

the long-term variability of the AdriSC model is well described by the change of sign of the main EOF components derived from Sea Surface Height (SSH) in the northern Ionian Sea. The main modes of variability of the Ionian-Adriatic exchanges are thus derived from the AdriSC ROMS 3-km monthly northern Ionian SSH over the 31-year period of the simulations, while their impact on the Adriatic Sea is extracted from the AdriSC ROMS 1-km results. All presented spatial EOFs are obtained via a covariance matrix and are normalized. The time series of the amplitudes associated with each eigenvalue in

the EOF are derived via the dot product of the data and the EOF spatial patterns, with the mean subtracted from each component time series.

The Ionian-Adriatic exchanges are also characterized with time series of both inward and outward deep-water mass transports along the Strait of Otranto (T6 transect) for the RCP 8.5 simulation, and from the Deep-Water Volume and SI within the DA subdomain for both historical ( $\sigma \geq 29.2$ kg/m³ criterion) and RCP 8.5 ( $\sigma \geq 29.09$ kg/m³ criterion)

simulations. Additionally, an animation of the Deep-Water Height in the southern Adriatic Sea for the RCP 8.5 simulation is provided (Movie S1).

## 3 Results

### 3.1 Bora Events

#### 3.1.1 Spatial extent and intensity

The spatial extent and intensity of the selected windstorms and their associated climate adjustments are first presented in Figs. 4 and 5. Between November and March in the historical simulation (Fig. 4), the horizontal wind speeds vary from north to south along the eastern Adriatic coast, with intense jets above 16 m/s separated by weaker speeds. This behavior is characteristic of the known Bora wake and gap jet dynamics (e.g., Jiang and Doyle, 2005; Gohm et al., 2008; Alpers et al., 2009; Signell et al., 2010), and the historical simulation can reproduce the known Trieste, Senj, Karlobag, and Sukošan main

Bora jets (see Fig. 4). For the rest of the year, in accordance with the known literature, some Bora jets are still present, but







**Figure 4. Historical monthly climatology of the spatial extent and intensity of the selected windstorms (≥ 13 m/s) defined as the monthly median wind speed at 10 m over the 31 years of the historical simulation.**



**Figure 5. Monthly climatology of the climate adjustments (in percent) associated with the selected Bora events (wind speeds at 10 m ≥ 13 m/s) and defined, over the 31 year of the simulations, as the difference between RCP 8.5 and historical monthly medians divided by historical monthly medians.**




their intensity is, on average, decreased (≤ 14 m/s). Consequently, using wind speeds ≥ 13 m/s to identify Bora winds is a
simple but efficient criterion, as, over the northern Adriatic Sea, windstorms are dominated by these events.

In terms of climate adjustments (Fig. 5), the far-future intensity of the Bora jets during DJFM is mostly reduced by about 5%
within the Kvarner Bay but increased by up to 15% (less than 5% on average) along the Trieste jet. Additionally, in October
and December, the intensity of the selected Bora winds is overall increased by 5 to 15%. Finally, for most months, an
alternation of strong reduction in intensity (up to 15% and 10% on average) and moderate increase (up to 15% but 5% on
average) along the known Bora jet locations can be seen in Fig. 5. Consequently, in the RCP 8.5 simulation, the locations of
the main Bora jets are most probably shifted in space while their intensity is overall reduced.

### 3.1.2 Monthly climatologies

For the selected Bora events, the impact of climate change on the air-sea dynamics is presented separately for the historical
and RCP 8.5 simulations as monthly climatologies of horizontal wind transports, accumulated surface buoyancy losses, and
total, latent, and sensible heat fluxes (Fig. 6), as well as air minus sea saturation specific humidity (SAT), relative humidity,
and freshwater fluxes (Fig. 7).

For both historical and RCP 8.5 simulations, the strongest horizontal wind transports (median value above $2800 \times 10^9$ m³/s)
occur between November and March. Compared to the historical results, the horizontal wind transports are overall largely
reduced in the RCP 8.5 simulation – i.e., between $20 \times 10^9$ m³/s in February and $281 \times 10^9$ m³/s in March – but increased in
January and August by about $170 \times 10^9$ m³/s and $140 \times 10^9$ m³/s, respectively. In terms of the most extreme wind transports,
defined as the 75th percentile, they are reduced in the RCP 8.5 simulation by up to $3071 \times 10^9$ m³/s in January and $1254 \times$
$10^9$ m³/s in February but are increased by up to $822 \times 10^9$ m³/s in September and $1325 \times 10^9$ m³/s in December.

The strongest accumulated surface buoyancy losses occur between September and March in both historical and RCP 8.5
simulations and can reach a monthly median of more than 0.030 m²/s² in December. In contrast with the horizontal wind
transports, the median RCP 8.5 accumulated surface buoyancy losses are overall increased by 0.004 m²/s² on average
compared to the historical simulation. This increase varies between 0.001 m²/s² in December and 0.017 m²/s² in November.
The extreme RCP 8.5 buoyancy losses, defined as the 75th percentile, are also increased all year long by 0.005 m²/s² on
average and by a maximum of 0.015 m²/s² in November.

Regarding the total, latent, and sensible monthly heat fluxes, for both RCP 8.5 and historical simulations, they reach their
maximum losses (median value above 150 W/m²) between September and March. Overall, compared to the historical results,
the RCP 8.5 total heat losses increase between 11 W/m² in February and 88 W/m² in November, with an average of 35 W/m²
between August and March, while the RCP 8.5 total heat gain decreases by about 17 W/m² on average between May and
July. In contrast, the RCP 8.5 latent heat losses increase all year long by at least 3 W/m² in April and up to 72 W/m² in
November (an average of 33 W/m²), while the RCP 8.5 sensible losses decrease by 6 W/m² on average most of the year
(except in March, October, and November, which have increased losses between 4 and 9 W/m²). In terms of extremes,



defined as the 25th percentile, the monthly RCP 8.5 latent heat losses are increased by 40 W/m² on average, and up to 42 W/m² in March and 52 W/m² in November.



**Figure 6. Monthly climatologies of the median, 25th and 75th percentiles of the horizontal wind transport at 10 m, the accumulated surface buoyancy loss and the total, latent and sensible heat fluxes defined over the 31 years of the historical and RCP 8.5 simulations for the selected Bora events.**





**Figure 7. Monthly climatologies of the median, 25th and 75th percentiles of the air minus sea saturation specific humidity, the relative humidity at 2 m and the fresh water flux defined over the 31 years of the historical and RCP 8.5 simulations for the selected Bora events (top panels). Climate adjustments (in percent) for the 8 variables used in the Bora event analyses presented as box plots during DJFM (bottom panel).**



For the remaining variables (Fig. 7), while the RCP 8.5 median monthly relative humidity changes by less than ±1% compared to the historical simulation, the median and extreme (represented by the 25th percentile) monthly losses of both air minus sea SAT and freshwater flux are expected to increase all year long by an average of 0.8 g/kg and $1.15 \times 10^{-8}$ m/s, respectively, and up to 1.5 g/kg in November and $1.6 \times 10^{-8}$ m/s in October, respectively.

### 3.1.3 Discussion

To summarize the results of the previous section, the monthly climate adjustments are presented as box plots (Fig. 7, bottom panel). These reveal that the median (and extreme, given by the 25th/75th percentile depending on the negative/positive sign of the median) differences between RCP 8.5 and historical results are -25% (-52%) for the horizontal wind transports at 10 m and -4% (-9%) for the sensible heat fluxes, but +15% (+30%) for the buoyancy losses, +8% (+20%) for the total heat fluxes, +17% (+24%) for the latent heat fluxes, +20% (+27%) for the air minus sea SAT, +1% (+2%) for the relative humidity at 2 m, and, finally, +17% (+24%) for the freshwater fluxes.

Consequently, a strong reduction of the intensity and spatial extent of the winter Bora winds is projected in the AdriSC RCP 8.5 far-future simulation. This confirms the findings of other regional and kilometer-scale atmospheric long-term models (e.g., Benetazzo et al., 2012; Androulidakis et al., 2015; Bonaldo et al., 2017; Belušić Vozila et al., 2019) and the 3-day-long AdriSC climate simulations (Denamiel et al., 2020a, 2020b). As previously seen in Denamiel et al. (2020b), the accumulated buoyancy losses, particularly the latent heat and freshwater losses, are strongly increased (by more than 15%) in the RCP 8.5 simulation, leading to strong cooling at the air-sea interface. In contrast with what was previously hypothesized in Denamiel et al. (2020b), the changes in relative humidity at 2 m are minor and cannot explain this increase. However, as the latent heat losses, the air minus sea SAT and the freshwater losses are projected to increase by at least 17%, the increase of the buoyancy losses under the RCP 8.5 conditions is mainly controlled by the increased evaporation and not by the decrease in Bora wind intensity and spatial extent. Given these results, in contrast with the findings of Parras-Berrocal et al. (2023), the NAddW formation is expected to be similar in far-future and present climates.

## 3.2 Adriatic Dense Water Dynamics

### 3.2.1 Spatial extent and intensity

The changes in the spatial extent and intensity of the dense water formation, propagation, and accumulation under the far-future extreme warming are first presented as spatial plots of the monthly historical DWH and their associated climate adjustments (Figs. 8 and 9). Importantly, the dynamical behavior of the SAP will be discussed in section 3.3 and will not be analyzed here.

In the historical simulation, the DWH ($\sigma \geq 29.2$ kg/m³ criterion) reaches a maximum within the northern Adriatic shelf (between 40 m in the shallow areas and 75 m in the deepest parts) and the Kvarner Bay (above 75 m) between December and April when the NAddW is formed and fills the formation sites (NA and KB subdomains). Within the deepest parts of the





**Figure 8. Historical monthly climatology of the median Dense Water Height (DWH; defined for $\sigma \geq 29.2$ kg/m3) over the 31 years of the historical simulation.**





**Figure 9. Monthly climatology of the climate adjustments (in percent) associated with the DWH and defined, over the 31 year of the simulations, as the difference between RCP 8.5 and historical monthly medians divided by historical monthly medians.**





Kvarner Bay (i.e., DKB subdomain), the DWH is still about 25 m in May and decreases to below 5 m in September when no
dense water is left in this accumulation site before December. In the Jabuka Pit accumulation site, the DWH peaks in
February and March (above 160 m) but remains above 125 m all year long in the deepest parts of the middle and western
areas of the pit. Along the Italian coast (i.e., the western side of the SAP), where the dense water is known to exit the
Adriatic basin, the DWH peaks in December and February (above 100 m) but varies between 5 m and 75 m the rest of the
year when exchanges of dense water occur between the Jabuka Pit and the SAP.

In terms of the climate adjustments (Fig. 9), compared to the historical simulation, the RCP 8.5 DWH presents changes
smaller than ±10% within the northern Adriatic, all year long except between April and May when it decreases up to 60% in
the area where the dense waters are known to exit the Kvarner Bay. However, during December, the RCP 8.5 DWH
increases by up to 100% in December. Within and off the Kvarner Bay the RCP 8.5 DWH decreases by up to 25% in April
but increases by up to 15% between July and September and, even, by up to more than 100% in October, when dense water
will still be present within the DKB subdomain in the RCP 8.5 simulation. It is thus expected that less NAddW is transported
off the Kvarner Bay between April and June. However, the biggest changes of RCP 8.5 DWH (up to ±100%) occur, all year
long, within the Jabuka Pit – where it decreases between 10% in June and up to 100% in February and March – and along
the western side of the SAP – where it increases between 10% in April and 100% in July but decreases by up to 100%
between January and March.

**3.2.2 Daily climatologies**

The impact of climate change on dense water dynamics is presented separately for the historical and RCP 8.5 simulations.
These are shown as monthly climatologies of Dense Water Volume (DWV; Fig. 10) and Stratification Index (SI; Fig. 11)
within the NA, KB, DKB, and JP subdomains (Fig. 1), as well as dense water transports (Fig. 12) along the transects T1 to
T5 (Fig. 1).

For both historical and RCP 8.5 simulations (Fig. 10), the largest DWV (defined for $\sigma \geq 29.2$ kg/m³ and $\sigma \geq 28.4$ kg/m³,
respectively) occurs between December and March (without much variability between the 25th and 75th percentiles) within
the NAddW formation sites (up to 5 and $3.7 \times 10^{11}$ m³ for the NA and KB subdomains, respectively) and within the DKB
accumulation site (up to $0.55 \times 10^{11}$ m³). However, for the JP subdomain, this occurs only between February and May (with
variability reaching 4 and $2.5 \times 10^{11}$ m³ for the historical and RCP 8.5 simulations, respectively). Compared to the historical
results, the RCP 8.5 DWV is overall identical within the NA, KB, and DKB subdomains but is reduced within the Jabuka Pit
by an average of less than $0.2 \times 10^{11}$ m³ (up to $1.0 \times 10^{11}$ m³ in March).

The highest values of the SI (Fig. 11) occur across all subdomains during summer (JAS), when the sea surface temperature is
at its maximum. During DJFM, when the NAddW is formed, the SI is always below 0.1 m²/s², except for the JP subdomain
where it varies between 0.2 and 0.7 m²/s² and 0.4 and 1.1 m²/s² in the historical and RCP 8.5 simulations, respectively.
Compared to the historical simulation, the RCP 8.5 SI median (and extreme, represented by the 75th percentile) gains reach,
during this period, 0.006 (0.01) m²/s² for NA, 0.005 (0.008) m²/s² for KB and DKB, and 0.2 (0.3) m²/s² for JP.





In terms of the NAddW transports, they mostly occur between December and May outward of the formation sites: up to 18.0 × 10⁶ kg/s along T1 in March and also in January for the RCP 8.5 simulation, up to 7.5 × 10⁶ kg/s along T2 in February and December for the historical and RCP 8.5 simulations, respectively, and up to nearly 35.0 × 10⁶ kg/s along T3 in March. In 315 both simulations, compared to T3, the transports towards the Ionian Sea are overall reduced along T4 by up to 10.0 × 10⁶ kg/s in March in the historical simulation and 15.0 × 10⁶ kg/s in December in the RCP 8.5 simulation. Finally, along T5, the transports are overall reduced under the RCP 8.5 conditions compared to the historical simulation, by 5.0 × 10⁶ kg/s and up to 9.0 × 10⁶ kg/s in March.

<figcaption>Figure 10. Daily climatologies of the median, 25th and 75th percentiles of the Dense Water Volume (DWV) defined over the NA, KB, DKB and JP subdomains for the 31 years of the historical (defined for $\sigma \geq 29.2$ kg/m³) and RCP 8.5 (defined for $\sigma \geq 28.4$ kg/m³) simulations.</figcaption>





**Figure 11. Daily climatologies of the median, 25th and 75th percentiles of the Stratification Index (SI) defined over the NA, KB, DKB and JP subdomains for the 31 years of the historical and RCP 8.5 simulations.**

### 3.2.3 Discussion

In contrast with the study by Parras-Berrocal et al. (2023), which used the same threshold to define the NAddW in the present and future climates, the presented results demonstrate that, under far-future RCP 8.5 conditions, not only are the median accumulated buoyancy losses expected to increase by 15% (Fig. 7, bottom panel), but the NAddW formation within the NA and KB subdomains and the accumulation within the DKB subdomain are also expected to remain identical. Indeed, there is no major change in median DWV (Fig. 13, top left panel). It should be noted that despite the increase between 48% and 78%, the median RCP 8.5 SI remains really small in these areas during DJFM (Fig. 12 and Fig. 13, top middle panel). Furthermore, for both historical and RCP 8.5 simulations, more NAddW is transported through T3 than through T1 and T2 combined, which means some NAddW is probably formed offshore of the NA and KB sites. Under the RCP 8.5 scenario, the



offshore formation of the NAddW is expected to increase as the median NAddW transports increase by 13% along T3, decrease by 9% along T2, and do not change along T1 (Fig. 13, top right panel).



**Figure 12. Daily climatologies of the median, 25th and 75th percentiles of the dense water transport defined along the T1 to T5 transects for the 31 years of the historical and RCP 8.5 simulations.**







**Figure 13. Climate adjustments (in percent) for the 3 variables used in the NAddW dynamics analyses presented as box plots during DJFM (top panels). Variability of the historical and RCP 8.5 pycnoclines defined between the median and 99th percentile of the PDA along the T3 to T5 transects and within the Jabuka Pit during DJFM (bottom panels).**

Within the Jabuka Pit accumulation site, the main cascading and accumulation of the NAddW shifts from March in the

historical simulation to December under the RCP 8.5 conditions (i.e., shift in peak reduction in dense water transports





between T3 and T4; Fig. 12). However, despite the increase in transports by 13% along T3 and their decrease by 19% along T4 (Fig. 13, top right panel), the RCP 8.5 DWV within the Jabuka Pit is reduced by 5% compared to the historical simulation (Fig. 13, top left panel). Comparing the pycnoclines along T3, JP, T4, and T5 (Fig. 13, bottom panels) reveals that there are more occurrences of the NAddW filling the full water column in the historical simulation than under the RCP 8.5 conditions
– i.e., above 100 m depth, the area between 29.2 kg/m³ and the 99th percentile (in blue) is twice as large as the area between 29.09 kg/m³ and the 99th percentile (in red). Over the Jabuka Pit, the increase by 45% of the RCP 8.5 SI is thus likely to hamper vertical mixing causing a diminution of the RCP 8.5 DWV despite the presence of RCP 8.5 NAddW varying between 28.5 and 29.2 kg/m³ at the bottom of the pit below 200 m depth (Fig. 13, bottom panels).

Finally, the reduction of the RCP 8.5 transports by 13% along T4 and 34% along T5 compared to historical conditions (Fig.
13, top left panel) can have two explanations. First, the accumulated NAddW from the Jabuka Pit cannot cascade within the deepest part of the SAP where the densities are higher and, hence, in contrast with the historical results, no strong density current is present. Second, the decrease in densities of the NAddW (from the Jabuka Pit to transect T5) due to the interaction with the ambient Adriatic waters is greater for the RCP 8.5 than the historical conditions, up to 0.3 and 0.2 kg/m³, respectively (Fig. 13, bottom panels). This suggests that, under the RCP 8.5 conditions, most of the NAddW exits the
Adriatic Sea along the western side of the SAP, which also explains the increase in DWH in this area between April and November (Fig. 9).

### 3.3 Ionian-Adriatic Exchanges

In this section, the SSH EOFs over the northern Ionian Sea are used to define the main modes of the Ionian-Adriatic exchanges. First, for both historical and RCP 8.5 simulations, the first SSH EOFs are linked to the interannual variability and
are not displayed here. Second, hereafter, both analysis and discussion of the results are presented together.

### 3.3.1 Bimodal Oscillation System (BiOS)

For the historical simulation, as described in Denamiel et al. (2022), the second SSH EOF, representing nearly 10% of the total signal (Fig. 14, top left panels), is linked to the Ionian-Adriatic Bimodal Oscillating System or BiOS (Gačić et al., 2010). In the present climate, the BiOS connects the quasi-decadal reversals of the Northern Ionian Gyre (NIG) circulation to
the salinity variability in the Adriatic Sea. During the anticyclonic phase of the NIG, the southern Adriatic Sea salinity decreases due to the advection of less-saline Modified Atlantic Water. During the cyclonic phase of the NIG, the salinity increases due to the advection of highly-saline Levantine/Eastern Mediterranean waters.

In the RCP 8.5 simulation, the BiOS signal appears as the third SSH EOF and represents only 8% of the total signal (Fig. 14, top right panels). From these results, the expected BiOS signal in the Ionian Sea for the RCP 8.5 scenario is similar in both
spatial pattern and time series to the one obtained for the historical simulation. Furthermore, under RCP 8.5 conditions, the correlations between the BiOS signal and the first salinity EOFs at 100 m depth and the bottom of the Adriatic Sea – representing 74% and 56% of the total signal, respectively (Fig. 14, middle panels) – reach more than 60% with a 2-year lag.





**Figure 14. Normalized spatial EOF components and associated time series of amplitude for both the historical and RCP 8.5 AdriSC ROMS 3-km Sea Surface Height (SSH) over the northern Ionian Sea (top panels). RCP 8.5 AdriSC ROMS 1-km salinity at 100 m and the bottom of the Adriatic Sea (middle panels). Time series of the daily vertical Potential Density Anomaly (PDA) and Dense Water Volume (DWV) in the JP subdomain for the RCP 8.5 simulation (bottom panels).**



Consequently, as these results are similar to those found for the historical simulation by Denamiel et al. (2022), the BiOS remains the main driver of Adriatic salinity variability under the presented PGW extreme warming scenario. Importantly, at

the bottom of the Adriatic Sea, the BiOS does not affect the deepest part of the SAP (i.e., the DA subdomain).

Finally, for the RCP 8.5 simulation, the Adriatic BiOS-driven salinity phases strongly impact the renewal of the Jabuka Pit collector site (Fig. 14, bottom panels): during the cyclonic phases (Fig. 13, in blue in the EOF time series of Adriatic salinity), both PDA and DWV increase (up to 29.2 kg/m³ and $5.0 \times 10^{11}$ m³, respectively), while during the anticyclonic phases (Fig. 14, in red in the EOF time series of Adriatic salinity), the PDA is largely decreased over the entire water column

(down to below 28.2 kg/m³) and no (or very little, below $0.5 \times 10^{11}$ m³) dense water is present in the Jabuka Pit collector.

### 3.3.2 Deep-water exchanges

In the RCP 8.5 simulation, the second SSH EOF – representing about 15% of the total signal over the Ionian Sea (Fig. 15, top right panels) – is a mode of Ionian-Adriatic exchanges that is not present in the historical simulation. The anti-correlation, without lag, between this new mode and the first EOF of the bottom PDA over the SAP – representing about

63% of the total PDA signal for depths below 800 m – reaches more than 80%. Consequently, this mode controls the deep-water content of the deepest part of the SAP (i.e., the DA subdomain) and, hence, the presence of AdDW under the PGW RCP 8.5 scenario. The first phase of this mode (Fig. 15, in blue in the EOF time series of SSH over the Ionian Sea) is present for 7 years at the beginning of the RCP 8.5 simulation. The second phase (Fig. 15, in red in the EOF time series of SSH over the Ionian Sea) lasts for 20 years, while during the last 4 years of the simulation, the mode reverts to the first phase.

The switch between the first and second phase of this new mode corresponds to the year 1994 in the historical simulation, which marks the shift in dominant deep-water source in the northern Ionian Sea from the Adriatic Sea to the Aegean Sea. In the historical simulation, this event – known as the Eastern Mediterranean Transient (EMT) – is characterized by massive dense water formation triggered by extreme heat losses and high salinity in the Aegean Sea during winter 1992-1993 (Roether et al., 1996, 2007; Klein et al., 1999; Velaoras et al., 2017). During the EMT, the northern Ionian Sea is filled with

very dense water from the Aegean Sea, and the intrusion of Adriatic-originated water into the Levantine basin is blocked (Akpinar et al., 2016; Li and Tanhua, 2020). As the PGW method uses the historical boundary forcing, the RCP 8.5 scenario presented in this study is also forced with the EMT signal modified with an extreme warming climatological change.

In the RCP 8.5 simulation, for $\sigma \geq 29.09$ kg/m³, no deep-water is present within the DA subdomain before the EMT. First, most of the NAddW (defined for $\sigma \geq 28.4$ kg/m³) is too light to cascade into the deepest part of the SAP (which has an

ambient density of about 29.0 kg/m3 before the EMT). Second, over the DA subdomain, the RCP 8.5 stratification index (SI) is at least multiplied by 7 compare to the historical conditions (Fig. 15, bottom panels) which highly hampers the far-future deep-ocean convection. This strongly contrast with the present climate conditions for which NAddW is known to partly transform into AdDW during deep-convection processes over the SAP. However, after 7 years of simulation, the EMT triggers new Ionian-Adriatic exchanges of deep-water, and the DA subdomain is filled with deep-water coming from the







**Figure 15. Normalized spatial EOF components and associated time series of amplitude for the RCP 8.5 AdriSC ROMS 3-km Sea Surface Height (SSH) over the northern Ionian Sea (top left panels). RCP 8.5 AdriSC ROMS 1-km bottom PDA, for depths greater than 800 m in the Adriatic Sea (top right panels). Time series of the daily deep-water transport along the T6 transect for the RCP 8.5 simulation (middle left panel). DWV and SI over the DA subdomain for the historical and RCP 8.5 simulations (middle right and bottom panels).**

northern Ionian Sea – i.e., inward transports and DWV within the DA subdomain up to $10.0 \times 10^6$ kg/s and $5.0 \times 10^{12}$ m³, respectively (see Movie S1 and Fig. 15, middle panels).





In the far-future simulation, the amount of dense water within the DA subdomain is thus controlled by the Ionian-Adriatic exchanges and is far lower than under the historical conditions (Fig. 15, middle right panel). Under present climate
conditions, the ventilation of the deepest part of the SAP by the NAddW is indeed known to occur regularly (Cardin et al., 2020). This is marked by strong peaks of DWV occurring every 3 to 5 years over the DA subdomain in the historical simulation and their absence in the RCP 8.5 simulation.

## 4 Conclusions

In this study, an analysis of the dynamics of Northern Adriatic dense Water (NAddW) and Adriatic Deep Water (AdDW) is
presented using the kilometer-scale atmosphere-ocean AdriSC model under the Pseudo Global Warming (PGW) assumption. Several findings differing from previous studies based on coarser Mediterranean climate models are revealed and summarized in Fig. 16 and as follows.

**Figure 16. Adriatic dense and deep- water far-future dynamics as seen by the kilometre-scale atmosphere-ocean AdriSC model**
**under the Pseudo-Global Warming assumption.**



Firstly, employing PGW forcing in the far-future simulation and thus imposing a strong vertical temperature gradient to the AdriSC ROMS 3-km initial and boundary conditions, clearly emphasizes the necessity to update thresholds for defining dense and deep waters to account for background density changes. In fact, this result is aligned with the supplementary study done by Parras-Berrocal et al. (2023), which demonstrates that the choice of threshold significantly influences the results of dense water formation within the Adriatic basin. However, this study reduces the NAddW threshold to 28.4 kg/m³ while the lowest threshold tested by Parras-Berrocal et al. (2023) is 28.8 kg/m³.

Secondly, analysis of air-sea interactions at NAddW generation sites demonstrates a 15% increase in winter surface accumulated buoyancy losses under extreme warming. This finding contrasts with previous studies that did not reproduce the changes in coastal evaporation which compensate for the well-known reduction in intensity and spatial extent of far-future Bora winds (found to be, on average, 25% in this study).

Thirdly, as a consequence of the first two points, the major finding of this study is that far-future NAddW formation under extreme warming is expected to remain similar to present conditions. However, in terms of NAddW transport and accumulation, the volume of dense water in the Jabuka Pit is projected to decrease due to higher stratification hampering the vertical mixing, while transports between the Jabuka Pit and the deepest part of the Southern Adriatic Pit (SAP) are expected to stop, as NAddW will be lighter than AdDW in the far-future.

Fourthly, the deepest part of the Adriatic basin is found to be mostly disconnected from the NAddW dynamics, and the far-future AdDW dynamics is expected to depend on density changes in the northern Ionian Sea. However, the presented RCP 8.5 AdDW results are strongly influenced by boundary conditions imposed on the AdriSC ROMS 3-km in the northern Ionian Sea and using the PGW methodology. Consequently, exchanges between the northern Ionian Sea and the deepest part of the SAP should be further investigated, for example, with kilometer-scale models capable of properly representing (and sampling) the Strait of Otranto and having open boundaries away from the Ionian Sea.

Finally, under the PGW assumption, this study finds that extreme warming is unlikely to affect the impact of the Bimodal Oscillation System (BiOS) on salinity variability in the Adriatic basin. However, similarly to the previous point, the impact of extreme warming on the BiOS itself (i.e., on the reversal of the Northern Ionian Gyre) is likely not captured by the AdriSC modeling suite due to the strong influence of boundary conditions imposed in the northern Ionian Sea.

Beyond the presented results, dense water formation in the Adriatic Sea plays a crucial role in sustaining a variety of marine species, ranging from deep-sea corals (Cushman-Roisin et al., 2001; Grubišić et al., 2014) to shallow-water mussels (Ballarin and Frizzo, 2004), sea urchins (Pais et al., 2012), and seagrasses (Boudouresque et al., 2009). Pelagic species such as European pilchards and anchovies also benefit indirectly from nutrient upwelling caused by dense water formation, which increases plankton availability (Santojanni et al., 2006a, b). Currently, the impact of climate change on these species has not been comprehensively studied in the Adriatic Sea but has only been addressed in global assessments (e.g., Wernberg et al., 2011; Doney et al., 2012; Bopp et al., 2013). As demonstrated in this study, which provides new insights into far-future NAddW dynamics, climate change impacts on Adriatic atmosphere-ocean processes are highly complex and necessitate the



use of high-resolution models. These processes also influence the biogeochemistry of the Adriatic basin, suggesting that this
study may pave the way for new assessments of the impact of extreme warming on ecology and fisheries in the Adriatic Sea.

**Appendix A**

**A.1 Atmospheric variables:**

| | | |
|---|---|---|
| $U_{10}$ | horizontal wind speed at 2 m | [m/s] |
| $U_a$ | horizontal wind speed at 2 m | [m/s] |
| $T_a$ | air temperature at 2 m | [°C] |
| $r_h$ | relative humidity at 2 m | [%] |
| $\rho_a$ | density of moist air at 2 m | [kg/m$^3$] |
| $\rho_w$ | density of freshwater | [kg/m$^3$] |
| $P_a$ | mean sea level pressure | [hPa] |
| $C_{pa} = 1004.67$ | specific heat capacity of the air | [J K$^{-1}$ kg$^{-1}$] |

**A.2 Ocean variables:**

| | | |
|---|---|---|
| $T_s$ | sea surface temperature | [°C] |
| $T_{sK}$ | sea surface temperature | [K] |
| $S_s$ | sea surface salinity | [PSU] |
| $g = 9.81$ | gravitational acceleration | [m/s$^2$] |
| $\rho_0 = 1025$ | reference density of seawater | [kg/m$^3$] |
| $C_{p0} = 3991.87$ | specific heat capacity of seawater | [J K$^{-1}$ kg$^{-1}$] |
| $\alpha$ | thermal expansion coefficient | [1/K] |



$\beta$        haline contraction coefficient        [1/PSU]

$\sigma$        potential density anomaly (PDA)        [kg/m³]

$\sigma_T$        PDA threshold for dense/deep waters        [kg/m³]

$\rho$        density of the seawater        [kg/m³]

### A.3 Horizontal wind transport:

The horizontal wind transport is defined as the integration of gale force winds (i.e., horizontal wind speeds at 10 m greater than 13 m/s) over the area where they occur in the northern Adriatic Sea (for latitudes above 43 °N). In this study, the monthly median of the horizontal wind transports is used as a proxy to quantify the impact of climate change on the intensity and spatial extent of the extreme Bora events.

$$T_{wind} = \iint U_{10}\left(U_{10} \geq 13\right) dxdy \qquad \text{[m}^3\text{/s]}$$

### A.4 Total, sensible, latent heat and freshwater fluxes:

For comparison purpose the air-sea fluxes are calculated in the same way than in Denamiel et al. (2020a, 2020b). In this study the monthly medians of the total, latent, sensible heat fluxes, relative humidity at 2 m, air minus sea saturation specific humidity (SAT) and fresh water fluxes are used to quantify the impact of climate change on the air-sea interactions over the northern Adriatic during extreme Bora events ( $U_{10} \geq 13$ ).

$Q_{swn}$        net shortwave radiations        [W/m²]

$Q_{lwd}$        downward longwave wave radiations        [W/m²]

$\in\; = 0.97$        infrared emissivity

$\sigma_{Stef-Bolt} = 5.670374419 \text{ x } 10^{-8}$ Stefan-Boltzmann constant        [W m⁻² K⁻⁴]

$Q_{lwn} = Q_{lwd} - \in \sigma_{Stef-Bolt} T_{sK}^4$        net longwave radiations        [W/m²]

$e_{sat}\left(T\right)$        saturation vapour pressure        [hPa]

$L\left(T\right) = 2501000 - 2370T$        latent heat of vaporization        [J/kg]

$C_H, C_E$        turbulent transfer coefficients



$$q_a \approx \frac{0.62197\left(0.01\, r_h\, e_{sat}\left(T_a\right)\right)}{p_a}$$ air saturation specific humidity at 2 m [kg/kg]

$$q_s \approx \frac{0.62197\left(0.98\, e_{sat}\left(T_s\right)\right)}{p_a}$$ sea surface saturation specific humidity [kg/kg]

$$Q_s = \rho_a C_H C_{pa} U_a \left(T_a - T_s\right)$$ sensible heat flux [W/m²]

$$Q_l = \rho_a C_E U_a L\left(T_s\right)\left(q_a - q_s\right)$$ latent heat flux [W/m²]

$$E_v = \frac{\rho_a}{\rho_w} C_E U_a \left(q_a - q_s\right)$$ evaporation rate [m/s]

$$P_r$$ precipitation rate [m/s]

$$Q_{Total} = Q_{swn} + Q_{lwn} + Q_s + Q_l$$ total heat fluxes [W/m²]

$$FWF = P_r - E_v$$ fresh water fluxes over the sea [m/s]

**A.5 Surface buoyancy fluxes and losses:**

In this study, the monthly accumulated surface buoyancy losses (BL) are used to quantify the air-sea fluxes over the northern Adriatic Sea during extreme Bora events ($U_{10} \geq 13$). The surface buoyancy fluxes (BF) are defined as in Parras-Berrocal et al. (2023) but the buoyancy losses (BL) are calculated monthly instead of over the DJFM period.

$$BF = g\left(\frac{\alpha Q_{Total}}{\rho_0 C_{pw}} + \beta S_S FWF\right)$$ surface buoyancy fluxes [m²/s³]

$$BL = -\int BF dt$$ monthly surface buoyancy losses [m²/s²]

**A.6 Dense and deep- water height and volume:**

The dense or deep- water height (DWH) is calculated over the vertical for a specific isopycnic surface ($\sigma_T$) assuming that the water column is stable. It is used to derive the dense/deep- water volume (DWV) quantifying the amount of dense/deep- water present within the specific subdomains chosen in this study.


$$\delta_{\sigma\sigma_T} = \begin{cases} 0 \text{ if } \sigma < \sigma_T \\ 1 \text{ if } \sigma \geq \sigma_T \end{cases}$$ Kronecker delta



$$DWH = \int \delta_{\sigma\sigma_T} dz \qquad \text{dense water height} \qquad [\text{m}]$$

$$DWV = \iint DWH dx dy \qquad \text{dense water volume} \qquad [\text{m}^3]$$

**A.7 Stratification Index:**

The Stratification Index (SI; Turner, 1973) is used in this study to assess the daily water column stratification (i.e., low values indicate a weak stratification and vice versa). For comparison purpose, the same vertical integration is used than in Parras-Berrocal et al. (2023) but the SI is defined as the median value over the specific subdomains chosen in this study and not over the whole Adriatic Sea.

$$N^2 = \frac{g}{\rho_0}\frac{\partial \rho}{\partial z} \qquad \text{with } N \text{ the Brunt–Väisälä frequency} \qquad [1/\text{s}^2]$$

$$SI = \int_0^h N^2 z dz \qquad \text{stratification index with } h = 650\,\text{m} \qquad [\text{m}^2/\text{s}^2]$$

**A.8 Dense or deep- water inward/outward transports along a vertical transect T:**

The dense/deep- water transports are calculated along the transects selected in this study and can be outward transports (i.e., exiting the Adriatic basin) or inward transports (i.e., entering the Adriatic basin).

$$U_N \qquad \text{ocean velocity normal to the transect T} \qquad [\text{m/s}]$$

$$x_T \qquad \text{distance along the transect T} \qquad [\text{m}]$$

$$M_{T\_outwards} = \iint \sigma(\sigma \geq \sigma_T) U_N (U_N \leq 0) dx_T dz \qquad [\text{kg/s}]$$

$$M_{T\_inwards} = \iint \sigma(\sigma \geq \sigma_T) U_N (U_N \geq 0) dx_T dz \qquad [\text{kg/s}]$$

**Code availability**

The code of the COAWST model as well as the ecFlow pre-processing scripts and the input data needed to re-run the AdriSC climate model can be obtained under the Open Science Framework (OSF) data repository (Denamiel, 2021) under the MIT license.



**Data availability**

The model results used to produce this article can be obtained under the Open Science Framework (OSF) FAIR data
repository (Denamiel, 2024) under the CC-By Attribution 4.0 International license.

**Video supplement**

Movie S1

**Author contribution**

CD designed and carried out the analyses presented in the study. CD developed the model code and performed the
simulations. CD prepared the manuscript with contributions from all co-authors.

**Competing interests**

The authors declare that they have no conflict of interest.

**Acknowledgments**

The computing and archive facilities used in this research were provided by the European Centre for Medium-range Weather
Forecasts (ECMWF) through national quota and the ECMWF Special Projects "The Adriatic decadal and inter-annual
oscillations: modelling component" and "Numerical modelling of the Adriatic-Ionian decadal and inter-annual oscillations:
from realistic simulations to process-oriented experiments". The research has been supported by the HORIZON EUROHPC
JU project ChEESE-2P (Grant 101093038).

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
