# Peer review of "A New Vision of the Adriatic Dense Water Future under Extreme Warming"

_EGUsphere, 2024_

## Author Comment (AC1)

**Response to Reviewer #1**

*The authors investigated the future evolution of dense and deep-water formation in the Adriatic Sea, as well as its further spread and accumulation at different locations in the sub-basin, following the pseudo global warming (PGW) approach under the RCP8.5 emissions scenario. For this purpose, the authors use kilometer-scale simulations performed with the Adriatic Sea and Coast (AdriSC) climate model. The AdriSC model has already been evaluated in previous work. By the end of the century, the authors find a NaddW formation similar to the present climate, while the AdDW formation is expected to decrease. The authors report some new findings that make the article very interesting. Indeed, the evolution and mechanism of deep-water formation is a central topic in oceanography, especially in the Mediterranean Sea. Its future changes could have implications for regional circulation, biogeochemistry and marine ecosystems. In my opinion, the main novelty of this work is that the authors have examined the future effects on the dynamics of the NaddW and the AdDW separately, thanks to the use of kilometer-scale simulations.*

*The manuscript is generally well written and organized, and the results are relevant to the scientific community, especially the Mediterranean community. However, I found some points that should be improved before its publication in OS.*

**Answer**: Thanks a lot for your interest in our study and for pointing out below several ways to improve the manuscript.

*1) Throughout the manuscript, the authors compare their results with those previously obtained by Parras-Berrocal et al. (2023) using a future projection performed with one of the coupled regional models participating in the Med-CORDEX initiative. However, I also miss a discussion with the results obtained by Soto-Navarro et al. (2020), who analyzed the future evolution of deep-water formation in the Adriatic Sea with an ensemble of fully coupled regional climate models from the Med-CORDEX. In this study, some models predict a decrease in deep water formation in the Adriatic, while others predict an increase. My advice is to try to position AdriSC within the Med-CORDEX ensemble. Does AdriSC project an average or an extreme behaviour with respect to the rest of the model simulations? The results in Soto-Navarro et al. 2020 (Table 9 and Figure S19) should help you to reinforce this point in the introduction and in the discussion.*

**Answer**: The authors fully agree with this suggestion and will compare their results with the study of Soto-Navarro et al. (2020), in particular with Table 9 and Figure S19 which present some specific results in the Adriatic Sea.

Introduction will thus be changed as follows (after line 58): "Soto-Navarro et al. (2020) analysed the future evolution of deep-water formation in the Adriatic Sea with the Med-CORDEX ensemble of fully coupled regional climate models in the Mediterranean Sea while Parras-Berrocal et al. (2023) studied the impact of climate change on dense water formation in the Eastern Mediterranean with one of the Med-CORDEX model. However, the results from both studies were averaged over the entire Adriatic Sea and the Med-CORDEX Regional Climate System Models (RCSMs) have coarse resolutions – 25 km in the atmosphere and about 15 km in the ocean – insufficient to represent the known NAddW dynamics accurately."

While the comparison of the AdriSC results with the Soto-Navarro et al. (2020) study will be added at the end of section 3.3.2 as follows: "Interestingly these results can be

compared to the study of Soto-Navarro et al. (2020) which found that most Med-CORDEX models project a reduction in the intensity of the deep convection events while one model projects an intensification of the convection in the Aegean Sea similar to what happened during the EMT in the 1990s. However, in the Aegean, Soto-Navarro et al. (2020) found that most Med-CORDEX models project a reduction of the DWF and, hence, the EMT-like situation, seen in the AdriSC model under the PGW assumption, is unlikely to occur."

*2) Regarding Figure 1, what criteria do you use to define the 5 subdomains (coloured polygons)? It should be stated somewhere in the text. Perhaps you could change the colour of the DKB polygon as it is quite similar to the colour used in NA. Could you include the meaning of MSL in the caption? The acronym MSL is not defined anywhere in the text.*

**Answer**: Firstly, it should be noted that, in the introduction, all the subdomains are already described: "NAddW formation occurs within the shallow Northern Adriatic shelf (Fig. 1, NA subdomain) and the Kvarner Bay (Fig. 1, KB subdomain) during strong surface heat and freshwater losses between December and March. The deepest part of the Kvarner Bay (Fig. 1, DKB subdomain) also acts as a dense water collector. NAddW then spreads southward along the western Adriatic coast and is partially collected within the Jabuka Pit (Fig. 1, JP subdomain) and the Southern Adriatic Pit (SAP; Fig. 1)." However, in order to further clarify the methodology in section 2.2.2, the following sentence will be added: "Dense Water Volume (DWV) and Stratification Index (SI) over 4 different subdomains (NA, KB, DKB, and JP; Fig. 1) covering the previously identified dense water formation and accumulation sites (i.e., identical to those used in Pranić et al.,2024) ...".

Secondly, the color of the DKB subdomain will be changed in order to have more contrasts with the other subdomains in the northern Adriatic. See figure below:

[Figure]

Finally, the meaning of MSL will be added in the caption of Figure 1. See new caption below:

**"Figure 1. Topo-bathymetry of the AdriSC climate model with the locations of the 5 subdomains (coloured polygons) and 6 transects (dotted black and white lines) used in the study. MSL stands for Mean Sea Level."**

3) One of my main concerns is that throughout the manuscript the authors give values for the differences between the RCP85 and historical periods. However, in most cases these differences are not easy to visualize in the figures because the authors show the historical and RCP8.5 periods separately. I strongly recommend that Figures 6, 7, 10, 11, 12 be redesigned to include a third column with the differences, or that both periods be combined in one column and the differences added in a second column. This would make the manuscript easier to follow.

**Answer**: the authors fully agree with the reviewer and will update all the climatological figures with a third column showing the differences between RCP 8.5 and historical conditions for the median and the extreme value (25th or 75th percentile depending on the sign of the studied variable). See new figures below:

[Figure]

**Figure 6. For the selected Bora events, monthly climatologies of the median, 25th and 75th percentiles of the horizontal wind transport at 10 m, the accumulated surface buoyancy loss and the total, latent and sensible heat fluxes defined over the 31 years for the historical, RCP 8.5 and RCP 8.5 minus historical conditions.**

[Figure]

**Figure 7. For the selected Bora events, monthly climatologies of the median, 25th and 75th percentiles of the air minus sea saturation specific humidity, the relative humidity at 2 m and the fresh water flux defined over the 31 years for the historical, RCP 8.5 and RCP 8.5 minus historical conditions (top panels). Climate adjustments (in percent) for the 8 variables used in the Bora event analyses presented as box plots during DJFM (bottom panel).**

[Figure]

**Figure 10. Daily climatologies of the median, 25th and 75th percentiles of the Dense Water Volume (DWV) defined over the NA, KB, DKB and JP subdomains for the 31 years of the historical (defined for $\sigma \geq 29.2$ kg/m³), RCP 8.5 (defined for $\sigma \geq 28.4$ kg/m³) and RCP 8.5 minus historical conditions.**

[Figure]

**Figure 11. Daily climatologies of the median, 25th and 75th percentiles of the Stratification Index (SI) defined over the NA, KB, DKB and JP subdomains for the 31 years of the historical, RCP 8.5 and RCP 8.5 minus historical conditions.**

[Figure]

**Figure 12. Daily climatologies of the median, 25th and 75th percentiles of the dense water transport defined along the T1 to T5 transects for the 31 years of the historical, RCP 8.5 and RCP 8.5 minus historical conditions.**

*4) Lines 333-336: What about the possibility of including a new subdomain located offshore NA and KB sites in the analysis?*

**Answer**: the authors acknowledge that this is the first time that the formation of dense water off the known sites (NA, KB) is analyzed but they believe that adding a subdomain would not greatly improve the manuscript as clearly the transports along the chosen transects (T1 to T3 which circle this formation site) as well as the spatial plots of monthly isopycnal depth allow to quantify/illustrate the dense water transport/formation there.

*5) In Figure 13, the SI appears to be calculated for the DJFM period. This is strange if the aim is to estimate the resistance of the water column to convection. Please consider computing it for a period before convection (for example in December or in autumn).*

**Answer**: the SI will be recalculated in December identically to Parras-Berrocal (2023). See figure new below:

[Figure]

**Figure 13. Climate adjustments (in percent) for the 3 variables used in the NAddW dynamics analyses presented as box plots during DJFM for DWV and mass transports and in December for SI (top panels). Variability of the historical and RCP 8.5 pycnoclines defined between the median and 99th percentile of the PDA along the T3 to T5 transects and within the Jabuka Pit during DJFM (bottom panels).**

*6) Please add letters to the figure panels at least in Figures 14 and 15. This will make the text easier to read and understand. Captions should be rewritten accordingly.*

**Answer**: the letters (a to d) will be added to Figures 14 and 15 and the captions will be changed. See new figures and captions below:

[Figure]

**Figure 14. a) Normalized spatial EOF components and associated time series of amplitude for both the historical and RCP 8.5 AdriSC ROMS 3-km Sea Surface Height (SSH) over the northern Ionian Sea. b) RCP 8.5 AdriSC ROMS 1-km salinity at 100 m and the bottom of the Adriatic Sea (middle panels). c) Time series of the daily vertical Potential Density Anomaly (PDA) and Dense Water Volume (DWV) in the JP subdomain for the RCP 8.5 simulation (bottom panels).**

[Figure]

**Figure 15. a) Normalized spatial EOF components and associated time series of amplitude for the RCP 8.5 AdriSC ROMS 3-km Sea Surface Height (SSH) over the northern Ionian Sea and the RCP 8.5 AdriSC ROMS 1-km bottom PDA, for depths greater than 800 m in the Adriatic Sea. b) Time series of the daily deep-water transport along the T6 transect for the RCP 8.5 simulation. c) DWV and d) SI over the DA subdomain for the historical and RCP 8.5 simulations.**

*7) Figure 16 is not easy to understand on its own. I would recommend following the numerical order used in the conclusions section to reorganize the figure, giving the same number to the processes involved in each conclusion. Otherwise, I would recommend removing Figure 16.*

**Answer**: the authors believe that Figure 16 is important as a visual summary of the study and will keep it. However, in order to facilitate the understanding of the figure, numbers and letters will be added to the figure and the following sentence will be added to the caption: "The numbers and letters correspond to the description of the different findings as described in the conclusions." See new figure and caption below:

[Figure]

**Figure 16. Visual summary of the study. Adriatic dense and deep- water far-future dynamics as seen by the kilometre-scale atmosphere-ocean AdriSC model under the Pseudo-Global Warming assumption. The numbers and letters correspond to the description of the different findings as described in the conclusions.**
* * *
*- L44: include density value of densest water based on previous literature.*

**Answer**: "(observed potential density anomalies up to 30.6 kg/m³; Raicich et al., 2013)" will be added.

*- L60: Regional Climate Model (RCM) → Regional Climate System Model (RCSM)*

**Answer**: will be done for the entire document.

*- Section 2.1.2: Could you indicate the length of the spin-up?*

**Answer**: the following sentence will be added: "As a rapid equilibrium is reached within the AdriSC ocean models (Pranić et al., 2021), a 2-month spin-up period allowing the atmosphere-ocean models to reach a steady state is used in both simulations."

*- In Figure 2, as well as in 3, 13, 14, I would recommend using Depth instead of Height.*

**Answer**: the choice between "depth" and "height" has been debated on several previously published articles and, to our understanding, as the presented values are

negative (hence, allowing for positive values inland), height should be used. Thus, height will be kept in this article.

*- L117: Please include a reference to the definition of extreme Bora events (i.e. gale-force winds).*

**Answer**: the reference from Belušić and Klaić (2004) will be added here.

*- L144: I would suggest replacing the term Dense Water Height (DWH) with isopycnal depth.*

**Answer**: will be replaced in the entire document.

*- In Figure 3, could you use the same colorbar range in both top panels? Another option is to include a sentence in the caption stating that different colorbar ranges are used to better visualize the figures.*

**Answer**: Using the same colorbar range is not really good as the difference in density between historical results and RCP 8.5 scenario is far too high. The following sentence will thus be added to the caption of Figure 3: "It should be noted that, in the colour plots, historical and RCP 8.5 results are presented with different extrema in density for a better visualization but with identical ranges (0.3 kg/m$^3$) to emphasis the increased stratification."

*- Line 153: "… summarized with box plots (Figure 13).."*

**Answer**: will be done. Reference to the other figures will also be added everywhere in the methodology section.

*- Figure 4: I would suggest including the vector field.*

**Answer**: the wind vector field will be added to figure 4. See below new figure.

[Figure]

**Figure 4. Historical monthly climatology of the spatial extent, intensity and direction (as vectors) of the selected windstorms (≥ 13 m/s) defined as the monthly median wind speed at 10 m over the 31 years of the historical simulation.**

*- Lines 212-217: This paragraph is hard to follow. In the text, the horizontal wind transport ranges from 20 to 2800 × 10⁹ m³/s, while in Figure 6 it ranges from 0 to 8 [10⁹ m³/s]. Please clarify the order of magnitude used.*

**Answer**: thanks a lot for spotting the mistake. The paragraph will be rewritten as follows: "For both historical and RCP 8.5 simulations, the strongest horizontal wind transports (median value above 2.8 × 10⁹ m³/s) occur between November and March. Compared to the historical results, the horizontal wind transports are overall reduced in the RCP 8.5

simulation – i.e., between 0.02 × 10⁹ m³/s in February and 0.28 × 10⁹ m³/s in March – but increased in January and August by about 0.17 × 10⁹ m³/s and 0.14 × 10⁹ m³/s, respectively. In terms of the most extreme wind transports, defined as the 75th percentile, they are reduced in the RCP 8.5 simulation by up to 3.07 × 10⁹ m³/s in January and 1.25 × 10⁹ m³/s in February but are increased by up to 0.82 × 10⁹ m³/s in September and 1.33 × 10⁹ m³/s in December."

*- Line 312: In terms of the Naddw transports (Figure 12). Sometimes in the text the authors do not refer to the figure they are talking about. Please check the whole manuscript.*

**Answer**: the figure numbers within the text will be checked and corrected when needed.

*- Line 327: In Parras-Berrocal et al. (2013), the authors consider NaddW and AdDW as Adriatic Deep Water without distinction due to the coarse resolution of the RCSM used.*

**Answer**: the sentence will be modified as follows: "In contrast with the study by Parras-Berrocal et al. (2023), which used the same threshold to define the NAddW in the present and future climates and considered NAddW and AdDW as deep-water without distinction due to the coarser resolution of the RCSM they used, …"

*- L345-346: " (i.e., shift in peak reduction in dense water transports between T3 and T4; Fig. 12)" I do not fully understand what the authors*

**Answer**: The sentence will be changed as follows: "(i.e., the maximum reduction in dense water transports between T3 and T4 which frame the JP is obtained in March under the historical conditions and in December under the RCP 8.5 conditions; Fig. 12)"

*- Line 351: 29.09 kg/m³ → 28.4 kg/m³?*

**Answer**: thanks for catching the mistake, will be changed.

*- Line 371: Modified Atlantic Water → Atlantic Water*

**Answer**: will be done.

*-References of this review not already cited by the authors:*

*Soto-Navarro, J., Jordá, G., Amores, A., Cabos, W., Somot, S., Se- vault, F., Macias, D., Djurdjevic, V., Sannino, G., Li, L., and Sein, D.: Evolution of Mediterranean Sea water properties un- der climate change scenarios in the Med-CORDEX ensemble, Clim. Dynam., 54, 2135–2165, https://doi.org/10.1007/s00382-019-05105-4, 2020.*

**Answer**: reference will be added.

---

## Author Response (AR1)

**Response to Reviewer #1**

*The authors investigated the future evolution of dense and deep-water formation in the Adriatic Sea, as well as its further spread and accumulation at different locations in the sub-basin, following the pseudo global warming (PGW) approach under the RCP8.5 emissions scenario. For this purpose, the authors use kilometer-scale simulations performed with the Adriatic Sea and Coast (AdriSC) climate model. The AdriSC model has already been evaluated in previous work. By the end of the century, the authors find a NaddW formation similar to the present climate, while the AdDW formation is expected to decrease. The authors report some new findings that make the article very interesting. Indeed, the evolution and mechanism of deep-water formation is a central topic in oceanography, especially in the Mediterranean Sea. Its future changes could have implications for regional circulation, biogeochemistry and marine ecosystems. In my opinion, the main novelty of this work is that the authors have examined the future effects on the dynamics of the NaddW and the AdDW separately, thanks to the use of kilometer-scale simulations.*

*The manuscript is generally well written and organized, and the results are relevant to the scientific community, especially the Mediterranean community. However, I found some points that should be improved before its publication in OS.*

**Answer**: Thanks a lot for your interest in our study and for pointing out below several ways to improve the manuscript.

*1) Throughout the manuscript, the authors compare their results with those previously obtained by Parras-Berrocal et al. (2023) using a future projection performed with one of the coupled regional models participating in the Med-CORDEX initiative. However, I also miss a discussion with the results obtained by Soto-Navarro et al. (2020), who analyzed the future evolution of deep-water formation in the Adriatic Sea with an ensemble of fully coupled regional climate models from the Med-CORDEX. In this study, some models predict a decrease in deep water formation in the Adriatic, while others predict an increase. My advice is to try to position AdriSC within the Med-CORDEX ensemble. Does AdriSC project an average or an extreme behaviour with respect to the rest of the model simulations? The results in Soto-Navarro et al. 2020 (Table 9 and Figure S19) should help you to reinforce this point in the introduction and in the discussion.*

**Answer**: The authors fully agree with this suggestion and are now comparing their results with the specific results in the Adriatic Sea of the study of Soto-Navarro et al. (2020).

Introduction has thus be changed as follows (after line 58): "Soto-Navarro et al. (2020) analysed the future evolution of deep-water formation in the Adriatic Sea with the Med-CORDEX ensemble of fully coupled regional climate models in the Mediterranean Sea while Parras-Berrocal et al. (2023) studied the impact of climate change on dense water formation in the Eastern Mediterranean with one of the Med-CORDEX model. However, the results from both studies were averaged over the entire Adriatic Sea and the Med-CORDEX Regional Climate System Models (RCSMs) have coarse resolutions – 25 km in the atmosphere and about 15 km in the ocean – insufficient to represent the known NAddW dynamics accurately."

While the comparison of the AdriSC results with the Soto-Navarro et al. (2020) study is added at the end of section 3.3.2 as follows: "Interestingly these results can be

compared to the study of Soto-Navarro et al. (2020) which found that most Med-CORDEX models project a reduction in the intensity of the deep convection events while one model projects an intensification of the convection in the Aegean Sea similar to what happened during the EMT in the 1990s. However, in the Aegean, Soto-Navarro et al. (2020) found that most Med-CORDEX models project a reduction of the DWF and, hence, the EMT-like situation, seen in the AdriSC model under the PGW assumption, is unlikely to occur."

*2) Regarding Figure 1, what criteria do you use to define the 5 subdomains (coloured polygons)? It should be stated somewhere in the text. Perhaps you could change the colour of the DKB polygon as it is quite similar to the colour used in NA. Could you include the meaning of MSL in the caption? The acronym MSL is not defined anywhere in the text.*

**Answer**: Firstly, in order to clarify the methodology and how the subdomains and transects were chosen the following paragraphs are added:

- in section 2.2.2: "The NA and KB subdomains are geographically defined. They cover the northern Adriatic shelf (with depths below 50m) and the Kvarner Bay (with depths ranging from 0 to 100m) and are previously identified dense water formation sites (e.g., Zore-Armanda, 1963; Pranić et al., 2024). Transects T1 and T2 are defined along the open boundary of these subdomains. The DKB and JP subdomains are defined for depths above 80m and 200m, respectively, and are accumulation sites. The dense waters generated in the Kvarner Bay, which is much deeper than the adjacent northern Adriatic shelf, are gravitationally attracted in the DKB while the JP is a well-researched dense water accumulation site (e.g., Zore-Armanda, 1963; Pranić et al., 2024). Transects T3 and T4 are located north and south of the JP subdomain with the aim to properly quantify and discriminate the NAddW transported southward from the one accumulated in the Jabuka Pit. Transect T5 is located north of the deepest part of the Adriatic (SAP) to quantify how much NAddW is reaching the middle Adriatic."
- In section 2.2.3: "The DA subdomain is defined for depths above 1000m and is encompassing the SAP identically to the study of Pranić et al. (2024)."

Secondly, the color of the DKB subdomain is changed in order to have more contrasts with the other subdomains in the northern Adriatic. See figure below:

[Figure]

Finally, the meaning of MSL is added in the caption of Figure 1. See new caption below:

**"Figure 1. Topo-bathymetry of the AdriSC climate model with the locations of the 5 subdomains (coloured polygons) and 6 transects (dotted black and white lines) used in the study. MSL stands for Mean Sea Level."**

*3) One of my main concerns is that throughout the manuscript the authors give values for the differences between the RCP85 and historical periods. However, in most cases these differences are not easy to visualize in the figures because the authors show the historical and RCP8.5 periods separately. I strongly recommend that Figures 6, 7, 10, 11, 12 be redesigned to include a third column with the differences, or that both periods be combined in one column and the differences added in a second column. This would make the manuscript easier to follow.*

**Answer**: the authors fully agree with the reviewer and have updated all the climatological figures with a third column showing the differences between RCP 8.5 and historical conditions for the median and the extreme value (25th or 75th percentile depending on the sign of the studied variable). See new figures below:

[Figure]

**Figure 6. For the selected Bora events, monthly climatologies of the median, 25th and 75th percentiles of the horizontal wind transport at 10 m, the accumulated surface buoyancy loss and the total, latent and sensible heat fluxes defined over the 31 years for the historical, RCP 8.5 and RCP 8.5 minus historical conditions.**

[Figure]

**Figure 7. For the selected Bora events, monthly climatologies of the median, 25th and 75th percentiles of the air minus sea saturation specific humidity, the relative humidity at 2 m and the fresh water flux defined over the 31 years for the historical, RCP 8.5 and RCP 8.5 minus historical conditions (top panels). Climate adjustments (in percent) for the 8 variables used in the Bora event analyses presented as box plots during DJFM (bottom panel).**

[Figure]

**Figure 10. Daily climatologies of the median, 25th and 75th percentiles of the Dense Water Volume (DWV) defined over the NA, KB, DKB and JP subdomains for the 31 years of the historical (defined for $\sigma \geq 29.2$ kg/m³), RCP 8.5 (defined for $\sigma \geq 28.4$ kg/m³) and RCP 8.5 minus historical conditions.**

[Figure]

**Figure 11. Daily climatologies of the median, 25th and 75th percentiles of the Stratification Index (SI) defined over the NA, KB, DKB and JP subdomains for the 31 years of the historical, RCP 8.5 and RCP 8.5 minus historical conditions.**

[Figure]

**Figure 12. Daily climatologies of the median, 25th and 75th percentiles of the dense water transport defined along the T1 to T5 transects for the 31 years of the historical, RCP 8.5 and RCP 8.5 minus historical conditions.**

*4) Lines 333-336: What about the possibility of including a new subdomain located offshore NA and KB sites in the analysis?*

**Answer**: the authors acknowledge that this is the first time that the formation of dense water off the known sites (NA, KB) is analyzed but they believe that adding a subdomain would not greatly improve the manuscript as clearly the transports along the chosen transects (T1 to T3 which circle this formation site) as well as the spatial plots of monthly isopycnal depth allow to quantify/illustrate the dense water transport/formation there.

*5) In Figure 13, the SI appears to be calculated for the DJFM period. This is strange if the aim is to estimate the resistance of the water column to convection. Please consider computing it for a period before convection (for example in December or in autumn).*

**Answer**: the SI is recalculated in December identically to Parras-Berrocal (2023). See figure new below:

[Figure]

**Figure 13. Climate adjustments (in percent) for the 3 variables used in the NAddW dynamics analyses presented as box plots during DJFM for DWV and mass transports and in December for SI (top panels). Variability of the historical and RCP 8.5 pycnoclines defined between the median and 99th percentile of the PDA along the T3 to T5 transects and within the Jabuka Pit during DJFM (bottom panels).**

*6) Please add letters to the figure panels at least in Figures 14 and 15. This will make the text easier to read and understand. Captions should be rewritten accordingly.*

**Answer**: the letters (a to d) are added to Figures 14 and 15 and the captions are changed. See new figures and captions below:

[Figure]

**Figure 14. a) Normalized spatial EOF components and associated time series of amplitude for both the historical and RCP 8.5 AdriSC ROMS 3-km Sea Surface Height (SSH) over the northern Ionian Sea. b) RCP 8.5 AdriSC ROMS 1-km salinity at 100 m and the bottom of the Adriatic Sea (middle panels). c) Time series of the daily vertical Potential Density Anomaly (PDA) and Dense Water Volume (DWV) in the JP subdomain for the RCP 8.5 simulation (bottom panels).**

[Figure]

**Figure 15. a) Normalized spatial EOF components and associated time series of amplitude for the RCP 8.5 AdriSC ROMS 3-km Sea Surface Height (SSH) over the northern Ionian Sea and the RCP 8.5 AdriSC ROMS 1-km bottom PDA, for depths greater than 800 m in the Adriatic Sea. b) Time series of the daily deep-water transport along the T6 transect for the RCP 8.5 simulation. c) DWV and d) SI over the DA subdomain for the historical and RCP 8.5 simulations.**

*7) Figure 16 is not easy to understand on its own. I would recommend following the numerical order used in the conclusions section to reorganize the figure, giving the same number to the processes involved in each conclusion. Otherwise, I would recommend removing Figure 16.*

**Answer**: the authors believe that Figure 16 is important as a visual summary of the study and are keeping it. However, in order to facilitate the understanding of the figure, numbers and letters are added to the figure and the following sentence is added to the caption: "The numbers and letters correspond to the description of the different findings as described in the conclusions." See new figure and caption below:

[Figure]

**Figure 16. Visual summary of the study. Adriatic dense and deep- water far-future dynamics as seen by the kilometre-scale atmosphere-ocean AdriSC model under the Pseudo-Global Warming assumption. The numbers and letters correspond to the description of the different findings as described in the conclusions.**
* * *
*- L44: include density value of densest water based on previous literature.*

**Answer**: "(observed potential density anomalies up to 30.6 kg/m³; Raicich et al., 2013)" is added.

*- L60: Regional Climate Model (RCM) → Regional Climate System Model (RCSM)*

**Answer**: done for the entire document.

*- Section 2.1.2: Could you indicate the length of the spin-up?*

**Answer**: the following sentence is added: "As a rapid equilibrium is reached within the AdriSC ocean models (Pranić et al., 2021), a 2-month spin-up period allowing the atmosphere-ocean models to reach a steady state is used in both simulations."

*- In Figure 2, as well as in 3, 13, 14, I would recommend using Depth instead of Height.*

**Answer**: the choice between "depth" and "height" has been debated on several previously published articles and, to our understanding, as the presented values are

negative (hence, allowing for positive values inland), height should be used. Thus, height is kept in this article.

*- L117: Please include a reference to the definition of extreme Bora events (i.e. gale-force winds).*

**Answer**: the reference from Belušić and Klaić (2004) is added here.

*- L144: I would suggest replacing the term Dense Water Height (DWH) with isopycnal depth.*

**Answer**: agreed and replaced in the entire document.

*- In Figure 3, could you use the same colorbar range in both top panels? Another option is to include a sentence in the caption stating that different colorbar ranges are used to better visualize the figures.*

**Answer**: Using the same colorbar range is not really good as the difference in density between historical results and RCP 8.5 scenario is far too high. The following sentence is thus added to the caption of Figure 3: "It should be noted that, in the colour plots, historical and RCP 8.5 results are presented with different extrema in density for a better visualization but with identical ranges (0.3 kg/m$^3$) to emphasis the increased stratification."

*- Line 153: "… summarized with box plots (Figure 13).."*

**Answer**: done. Reference to the other figures is also added everywhere in the methodology section.

*- Figure 4: I would suggest including the vector field.*

**Answer**: the wind vector field is added to figure 4. See below new figure.

[Figure]

**Figure 4. Historical monthly climatology of the spatial extent, intensity and direction (as vectors) of the selected windstorms (≥ 13 m/s) defined as the monthly median wind speed at 10 m over the 31 years of the historical simulation.**

*- Lines 212-217: This paragraph is hard to follow. In the text, the horizontal wind transport ranges from 20 to 2800 × 10⁹ m³/s, while in Figure 6 it ranges from 0 to 8 [10⁹ m³/s]. Please clarify the order of magnitude used.*

**Answer**: thanks a lot for spotting the mistake. The paragraph is rewritten as follows: "For both historical and RCP 8.5 simulations, the strongest horizontal wind transports (median value above 2.8 × 10⁹ m³/s) occur between November and March. Compared to the historical results, the horizontal wind transports are overall reduced in the RCP 8.5

simulation – i.e., between 0.02 × 10⁹ m³/s in February and 0.28 × 10⁹ m³/s in March – but increased in January and August by about 0.17 × 10⁹ m³/s and 0.14 × 10⁹ m³/s, respectively. In terms of the most extreme wind transports, defined as the 75th percentile, they are reduced in the RCP 8.5 simulation by up to 3.07 × 10⁹ m³/s in January and 1.25 × 10⁹ m³/s in February but are increased by up to 0.82 × 10⁹ m³/s in September and 1.33 × 10⁹ m³/s in December."

*- Line 312: In terms of the Naddw transports (Figure 12). Sometimes in the text the authors do not refer to the figure they are talking about. Please check the whole manuscript.*

**Answer**: the figure numbers within the text have been checked and corrected when needed.

*- Line 327: In Parras-Berrocal et al. (2013), the authors consider NaddW and AdDW as Adriatic Deep Water without distinction due to the coarse resolution of the RCSM used.*

**Answer**: the sentence is modified as follows: "In contrast with the study by Parras-Berrocal et al. (2023), which used the same threshold to define the NAddW in the present and future climates and considered NAddW and AdDW as deep-water without distinction due to the coarser resolution of the RCSM they used, …"

*- L345-346: " (i.e., shift in peak reduction in dense water transports between T3 and T4; Fig. 12)" I do not fully understand what the authors*

**Answer**: The sentence is changed as follows: "(i.e., the maximum reduction in dense water transports between T3 and T4 which frame the JP is obtained in March under the historical conditions and in December under the RCP 8.5 conditions; Fig. 12)"

*- Line 351: 29.09 kg/m³ → 28.4 kg/m³?*

**Answer**: changed, thanks for catching the mistake.

*- Line 371: Modified Atlantic Water → Atlantic Water*

**Answer**: done.

*-References of this review not already cited by the authors:*

*Soto-Navarro, J., Jordá, G., Amores, A., Cabos, W., Somot, S., Se- vault, F., Macias, D., Djurdjevic, V., Sannino, G., Li, L., and Sein, D.: Evolution of Mediterranean Sea water properties un- der climate change scenarios in the Med-CORDEX ensemble, Clim. Dynam., 54, 2135–2165, https://doi.org/10.1007/s00382-019-05105-4, 2020.*

**Response to Reviewer #2**

*A kilometer-scale atmosphere-ocean model is used to assess the impact of a far-future extreme warming scenario on the formation, spreading and accumulation of the dense water in the Adriatic Sea.*

*The text is well written and results are presented clearly. I recommend minor corrections:*

**Answer**: Thanks a lot for your interest in our study and for pointing out below ways to improve the manuscript.

*105 : Please better explain the PGW approach and its consequences on the model behavior, as it is a key point for the paper.*

*455 : The choice of the boundary conditions and its implication on the exchange at the southern border are not well described. Could authors further discuss this point.*

**Answer**: The authors agree that the PGW method can be better described and that the consequences of having boundary conditions within the northern Ionian Sea must also be explained. The following paragraph is added in section 2.1.2: "For the atmosphere, the ERA-Interim air temperature, relative humidity, and horizontal wind velocities, defined on 37 atmospheric pressure levels, are modified between 1000 and 70 hPa with the climatological changes ΔT, ΔRH, ΔU, and ΔV, respectively. These changes are derived from the RCP 8.5 scenario of the LMDZ4-NEMOMED8 RCSM by subtracting the atmospheric results of the 1987–2017 period from those of the 2070–2100 period, producing 6-hourly three-dimensional climatological changes for the 366 days of the year. These new forcings are then used to provide the boundary and initial conditions for the WRF 15-km model in the PGW simulation. For the ocean, the MEDSEA ocean temperature, salinity, and currents, defined on 72 unevenly spaced vertical levels, are modified with the climatological changes ΔT ocean, ΔS ocean, ΔU ocean, and ΔV ocean, respectively. These changes are also derived from the RCP 8.5 scenario of the LMDZ4-NEMOMED8 RCSM to produce three-dimensional daily climatological changes for the 366 days of the year. These forcings are then used to provide the boundary and initial conditions for the ROMS 3-km model in the PGW simulation. In other words, the same climatological changes are used to modify the boundary conditions for each simulated year of the reanalysis period and the PGW simulations "inherit" the synoptic environment and weather/ocean conditions from the atmosphere-ocean reanalyses at the lateral boundaries. As a result, the main limitation of this methodology, compared to traditional downscaling techniques (Brogli et al., 2023), is that potential changes in intra-annual and interannual variability may be missed in the PGW projections. Additionally, in the presented RCP 8.5 simulation, due to the location of the AdriSC ROMS 3-km boundary conditions, the northern Ionian ocean dynamics may be more influenced by the MEDSEA reanalysis than by the projected climatic changes."

---

## Referee Report (RR1)

In this work, the authors present the evolution of dense and deep-water formation and its distribution across various locations in the Adriatic Sea in a well-written manner, with results that are of great interest to the research community.

The manuscript has already passed the first revision, and after reviewing the reviewers' suggestions, comments, and the authors' responses, I find the paper to be well-written with a highly interesting approach. The authors have thoroughly addressed all reviewer comments, resulting in a more robust and cohesive paper. Therefore, I recommend the acceptance of this paper.

I would suggest a few technical (but not crucial) improvements, such as labeling each subfigure in all figures, as reviewers have recommended for Figures 14 and 15. This would ensure a consistent presentation throughout the manuscript and enhance readability for the audience.

Additionally, the authors could also reference the work of Mihanović et al. (2013), *"Exceptional dense water formation on the Adriatic shelf in the winter of 2012,"* along with Raicich et al. (2013), as these studies are relevant to the densest water measurements in the area.

---

## Author Response (AR2)

Dear Reviewers and Editor,

Thank you very much for your valuable reviews and comments that truly help to improve the manuscript. As suggested by the third reviewer, subplot letters have been added to all figures and the Milhanović et al. (2013) reference has also been added.

Some minor changes (that can be tracked in the track change versions) have also been made in the txt in order to update the figure numbers with the subplot letters and to fix some minor formatting. Additionally, the Pranić et al. (2024) reference has been updated as the article is now published.